Comparison of the synergistic effect of lipid nanobubbles and SonoVue microbubbles for high intensity focused ultrasound thermal ablation of tumors

Yao Yuanzhi 1
Yang Ke 2
Cao Yang 1
Zhou Xuan 3
Xu Jinshun 1
Liu Jianxin 1
Wang Qi 4
Wang Zhigang 1 wzg62942443@163.com
Wang Dong 1 2 wangketongtong@163.com
1 Chongqing Key Laboratory of Ultrasound Molecular Imaging, the Second Affiliated Hospital of Chongqing Medical University, Chongqing Medical University , Chongqing , China
2 Department of Ultrasound, Children’s Hospital of Chongqing Medical University, Chongqing Medical University , Chongqing , China
3 Department of Emergency, Chinese PLA General Hospital , Beijing , China
4 Institute of Ultrasound Engineering in Medical of Chongqing Medical University, Chongqing Medical University , Chongqing , China
Huisman Henkjan
Electronic publication date: 2016 Feb 22
Publication date: 2016
Volume: 4
Electronic Location ID: e1716
Received 2015 Oct 7; Accepted 2016 Jan 30
Copyright: ©2016 Yao et al.
Copyright year: 2016
Copyright holder: Yao et al.
License: This is an open access article distributed under the terms of the Creative Commons Attribution License, which permits unrestricted use, distribution, reproduction and adaptation in any medium and for any purpose provided that it is properly attributed. For attribution, the original author(s), title, publication source (PeerJ) and either DOI or URL of the article must be cited.
License URL: https://creativecommons.org/licenses/by/4.0/

Keywords: Nanobubbles, Microbubbles, High intensity focused ultrasound, Tumor therapy

Funding: National Science Foundation of China 81371579 81571688 81401503 The Natural Science Foundation of Chongqing cstc2013jcyjA10002 Postdoctoral Project of Chongqing Xm2014054 Chongqing Municipal Health Bureau 2013-1-024 Program for Innovation Team Building at Institutions of Higher Education in Chongqing KJTD201303 This work was supported by the National Science Foundation of China (81371579, 81571688, 81401503), the Natural Science Foundation of Chongqing (cstc2013jcyjA10002), the Postdoctoral Project of Chongqing (Xm2014054), the Key project of Chongqing Municipal Health Bureau (2013-1-024) and the Project Supported by Program for Innovation Team Building at Institutions of Higher Education in Chongqing (KJTD201303). The funders had no role in study design, data collection and analysis, decision to publish, or preparation of the manuscript.

==============================
Microbubbles (MBs) are considered as an important enhancer for high intensity focused ultrasound (HIFU) treatment of benign or malignant tumors. Recently, different sizes of gas-filled bubbles have been investigated to improve the therapeutic efficiency of HIFU thermal ablation and reduce side effects associated with ultrasound power and irradiation time. However, nanobubbles (NBs) as an ultrasound contrast agent for synergistic therapy of HIFU thermal ablation remain controversial due to their small nano-size in diameter. In this study, phospholipid-shell and gas-core NBs with a narrow size range of 500–600 nm were developed. The synergistic effect of NBs for HIFU thermal ablation was carefully studied both in excised bovine livers and in breast tumor models of rabbits, and made a critical comparison with that of commercial SonoVue microbubbles (SonoVue MBs). In addition, the pathological changes of the targeted area in tumor tissue after HIFU ablation were further investigated. Phosphate buffer saline (PBS) was used as the control. Under the same HIFU parameters, the quantitative echo intensity of B-mode ultrasound image and the volume of coagulative necrosis in lipid NBs groups were significantly higher and larger than that in PBS groups, but could not be demonstrated a difference to that in SonoVue MBs groups both ex vivo and in vivo. These results showed that the synergistic effect of lipid NBs for HIFU thermal ablation were similar with that of SonoVue MBs, and further indicate that lipid NBs could potentially become an enhancer for HIFU thermal ablation of tumors.

Introduction

High intensity focused ultrasound (HIFU) has been gained widespread attention in research and application of tumor treatment (Hassanuddin et al., 2014; Peek et al., 2015; Zhang et al., 2011). Owing to the true noninvasiveness, availability and economic benefit in clinical practice, HIFU has achieved rapid development in the treatment of benign and malignant solid tumors in breast, prostate, liver and pancreas tissue in the past years (Cavallo Marincola et al., 2015; Uchida et al., 2012; Kazarian et al., 2008). Even more exciting is that HIFU thermal ablation therapy has already been FDA-approved for treating uterine fibroids in United States (Hesley, Gorny & Woodrum, 2013). However, although the development of HIFU for tumor treatment was quite inspiring, HIFU is still restricted by its intrinsic limitations for large tumors (Zhou, 2011). The HIFU treatment time is currently on the order of hours and lesions formed by single HIFU exposure are fairly small (several to dozen mm3). For example, to achieve a large volume of tumors destruction hundreds of HIFU exposure, hours of treatment and/ or higher ultrasound power is required (Fischer, Gedroyc & Jolesz, 2010). However, side effects including skin burns and unintended heating to healthy tissue are inevitable as a result of long treatment time and high ultrasound power required for continuous lesion formation. Therefore, in order to overcome its intrinsic limitations, one strategy is to improve the therapeutic transducer based on multiple elements transducer using fast electronic-steering phased array transducer, which belongs to the HIFU engineering field (Ellens et al., 2015). Another strategy to accelerate the therapeutic efficiency of HIFU is to introduce the enhancer into the targeted region during HIFU exposure (Moyer et al., 2015; Sun et al., 2012; Hamano et al., 2014; Ma et al., 2014).

At present, different sizes of gas-filled bubbles were introduced into the targeted region to enhance the therapeutic efficiency of HIFU thermal ablation and reduce side effects through changing the acoustic property of the targeted tissue, resulting in the ultrasound energy accumulation in the intended target region to damage the tissue (Hamano et al., 2014; Ma et al., 2014; Zhang et al., 2014; Zhou et al., 2015). Microbubbles (MBs) are well known to be an important enhancer for synergistically and extensively accelerating the lesion formation of ultrasound-mediated heating and cavitation activity during HIFU treatment procedure (Luo et al., 2006; Peng et al., 2012). In addition, HIFU has the potential of inducing anti-tumor immune response, and simultaneously bubbles could remarkably improve the anti-tumor immune response (Wu et al., 2004; Liu et al., 2012). Bubbles in the high acoustic pressures will experience nonlinear oscillation known as inertial cavitation, and radiate out of higher frequencies ultrasound with massive energy, which are more readily absorbed by tissues and availably converted into heat to damage the tumor cells (Coussios et al., 2007). MBs serving as cavitation nuclei in the targeted area could lower cavitation thresholds and enhance tissue heating during HIFU treatment (Tran et al., 2005; McDannold, Vykhodtseva & Hynynen, 2006). However, owing to MBs rapidly disappearing from circulation, multiple infusions during HIFU treatment are required. Additionally, because of MBs too large in size, multiple infusions in a time are risk of gas embolism. In order to overcome these limitatios of MBs, phase-shift nanodroplets are being now developed as a cavitation nucleation agent to enhance HIFU-mediated thermal ablation (Kopechek et al., 2014; Zhang et al., 2011). Many of the liquid perfluorocarbon droplets in previous studies composed of relatively high boiling-point perfluorocarbons in order to achieve stability, but which required more acoustic energy to induce vaporization (Rapoport et al., 2011; Kopechek et al., 2014). Meanwhile, the generated gas bubbles from liquid perfluorocarbon droplets will easily coalesce into large ones to occlude arterioles, causing unwanted arterial occlusion, ischemia and potentially tissue infarction. Hence, the safety of these liquid perfluorocarbon droplets in the body needs further investigation.

With the development of nanotechnology, nanobubbles (NBs) with various shells (protein, polymers and phospholipids) have increasingly attracted more attention in ultrasound molecular imaging and disease therapy (Huang et al., 2013). Based on several previous studies (Yin et al., 2012; Fan et al., 2013), phospholipid-shell and gas-core NBs have shown optimal imaging abilities, high stability and easy penetrability of tumor vessels. Consequently, NBs may potentially become a good alternative to MBs and/or liquid perfluorocarbon droplets in the ultrasound therapy. Moreover, current research on NBs mostly focuses on ultrasound molecular imaging and drug or gene carriers (Yin et al., 2012; Xie et al., 2015; Cavalli, Bisazza & Lembo, 2013). The research of NBs in synergistic HIFU thermal ablation is still in its initial stages (Zhang et al., 2014; Wang et al., 2012). In this study, we would like to further introduce lipid NBs and SonoVue MBs into HIFU treatment procedure to evaluate the effect of lipid NBs for synergistic HIFU ablation.

Herein, we introduced lipid NBs and Sulfur hexafluoride MBs (SonoVue, routinely used in clinic) to enhance HIFU thermal ablation and explore the synergistic effect of lipid NBs and SonoVue MBs in excised bovine liver and in vivo breast tumor models of rabbits during HIFU treatment procedure. NBs were successfully fabricated with phospholipids shell materials similar to those of sulfur hexafluoride microbubbles. Firstly, we compared the echo intensity of B-mode ultraosund image and the volume of the necrotic tissues induced by diverse HIFU parameters combined with lipid NBs or SonoVue MBs in ex vivo bovine liver. Then, we equally made a comparison in rabbits VX2 breast tumor models, and detected the pathological change after HIFU thermal ablation using hematoxylin-eosin (HE) staining and transmission electron microscope (TEM) and examined the proliferation of tumor cells using immunohistochemistry.

Materials and Methods

Preparation of NBs and MBs

The lipid NBs were fabricated as described previously (Wang et al., 2010). Briefly, 5 mg 1,2-dipalmitoyl-sn-glycero-3-phosphocholine (DPPC) and 2 mg 1,2-dipalmitoyl-sn-glycero-3-phosphoethanolmine (DPPE) were mixed in a vial (actual volume 1.5 ml), then 450 µl phosphate buffer saline (PBS) and 50 µl glycerol were added to prepare suspension, and then the suspension was incubated in a water bath at 45 °C for 30 min. The vial was capped with a rubber cap, and then the air in the vial was replaced with perfluoropropane (C3F8) gas. Finally, the suspension was mechanically vibrated at 4,000 rpm for 90 s. The purified lipid NBs were separated from the prepared bubbles with various diameters using a low-speed centrifugation (300 rpm, 3 min) and a higher-speed centrifugation (800 rpm, 5 min) for three times. The purified lipid NBs were finally resuspended in 3 ml PBS and stored at 4 °C for further use.

The MBs used in this study were commercial sulfur hexafluoride microbubbles (SonoVue, Bracco, Italy), which were approved for routine use in clinical ultrasonography in China. 5 ml sterile saline was added into the vial before use to form a liquid solution by vigorous vibration.

The particle sizes of lipid NBs and SonoVue MBs were measured using dynamic light scattering (Zeta SIZER3000HS; Malvern, UK). Fresh bubbles were prepared for each experiment. Meanwhile, the concentration of lipid NBs and SonoVue MBs were also counted with globulimeter (Yangling, Jiangsu, China) according to the calculation guidelines.

Animal models

All the rabbits in this study, weighing about 2.0∼2.5 kg, were purchased from the Animal Center of Chongqing Medical University. The VX2 tumor was obtained from the laboratory of Ultrasound Engineering Institute of Chongqing Medical University. All animal experimental procedures were approved by the Animal Ethics Committee of Chongqing Medical University (Approval Number SYXK (Chongqing) 2012-0001).

Under sterile conditions, the VX2 tumor tissue was cut into small pieces (about 0.5∼1.0 mm) using ophthalmic scissors and then cultivated in the mammary tissue of rabbits underneath the second bilateral nipples (Sun et al., 2012). To prevent infection, 800,000 units of penicillin were intramuscularly injected for three days.

Ex vivo excised bovine liver ablation by HIFU

The JC-200 focused ultrasound system (Chongqing Haifu Technology, Chongqing, China) was used as described previously (Wang et al., 2012). The system mainly consists of therapeutic ultrasound unit, diagnostic ultrasound unit and a central processing system. The therapeutic transducer has a focal length of 140 mm, a diameter of 220 mm and a working frequency of 0.94 MHz. The focal region is 9.8 mm along the beam axis and 1.3 mm in the transverse direction. The diagnostic transducer with center frequencies of 3.5–5 MHz was installed in the center of the therapeutic transducer and moved together to guide and monitor the treatment procedure in real time. The integrated transducers are submerged in degassed water.

Fresh bovine liver tissues were purchased from local abattoir and used within 12 h after slaughter. The liver was sliced into 12 cm × 6 cm × 6 cm in size, and immersed into a normal saline and degassed at 37 °C for 30 min. Lipid NBs (200 µl, 1.0 × 105 bubbles/ml) were directly injected into the targeted area using a syringe and monitored by the diagnostic ultrasound unit. After injection, HIFU thermal ablation was immediately performed on the injection site with diverse HIFU parameters (120 W for 5 s; 150 W for 5 s; 180 W for 5 s). Similarly, SonoVue MBs (200 µl, 1.0 × 105 bubbles/ml) and PBS (200 µl, pH = 7.4) were employed for comparison under the same HIFU parameters. Before and after HIFU exposure, the echo intensity of B-mode ultrasound image within region of interest (ROI) was automatically calculated using Gray Val 1.0 Software affiliated to HIFU system. Each group was repeatedly carried out eight times. The mean echo intensity for each ROI was calculated separately in this program. The average value of the eight ROIs echo intensities were adopted as the mean echo intensity of the group. After HIFU thermal ablation, the length, width and depth of necrotic tissues were measured to calculate their volumes in the bovine liver using the following formula: V = π∕6 × L × W × D (L: length, W: width, D: depth).

In vivo HIFU thermal ablation of rabbit breast VX2 tumors

Three weeks after implantation of tumors, the rabbit breast VX2 tumors (about 10 mm in diameter) were carefully depilated with 8% Na2S again. After anesthetization using 3% pentobarbital sodium (30 mg/kg), the rabbits were placed on the treatment couch in a prone position, and the tumor tissues were completely immersed into the degassed water. The targeted site of HIFU thermal ablation was guided and monitored using the diagnostic transducer before HIFU exposure. Forty-eight rabbits were randomly divided into three groups: HIFU combined with PBS (PBS group), HIFU combined with SonoVue MBs (MBs group) and HIFU combined with lipid NBs (NBs group), respectively. Each group had sixteen rabbits. In these three groups, the rabbits received an intravenous injection of PBS, SonoVue MBs and lipid NBs solution (0.2 ml/kg, 1.0 × 108 bubbles/ml) respectively, and received HIFU exposure after 15 s. During whole experiment process, single “ablated-dot” mode was employed and HIFU exposure parameters were kept the same with an acoustic power at 150 W and exposure time for 5 s. Before and after HIFU exposure, the echo intensity of the B-mode ultrasound image within ROI was also calculated using Gray Val 1.0 Software.

Three days after HIFU exposure, the animals were euthanized with intravenous 3% pentobarbital (1.5 ml/kg) and each breast tumor tissue was removed immediately for macroscopic and microscopic examinations. The tissue were cut into 2∼3 mm-thick slices to calculate the volume of the necrotic tissues exposed by HIFU according to the above equation. And then, the tissues were isolated, fixed with formalin, embedded in paraffin. The sections were subsequently stained with hematoxylin and eosin (HE) for pathological examination. In addition, about 1 mm3 necrotic tissue was used to evaluate the ultra-structure changes of cells using transmission electron microscope (TEM).

The immunohistochemical staining was used to detect the tumor cell proliferation (Luo et al., 2007). After being deparaffined in xylene and rehydrated, the sections were blocked with goat serum for 20 min at room temperature, and then incubated with mouse anti-proliferating cell nuclear antigen (PCNA) monoclonal antibody (diluted 1:500, Boster, Wuhan, China). After rinsing with PBS, the sections were incubated with biotinylated secondary antibody, followed by avidin-biotin peroxidase complex treatment, then counterstained with hematoxylin for 2 min. The PCNA positive cells showed brownish. The PCNA positive index (%) = the number of PCNA positive cells/total number of cells observed. Five random areas (400 × magnification) in each slide were observed.

Statistical analysis

SPSS 20.0 software (SPSS Co., Chicago, USA) was used to analyze the data. All data were expressed as the mean ± standard deviation. Group differences were analyzed with an independent sample t-test and intragroup comparison was performed using analysis of variance. P value less than 0.05 was considered as a significant difference.

Results

Characterization of NBs and MBs

Lipid NBs were successfully prepared with high stability and without morphologic change in PBS solution at room temperature for more than three days. The appearance of lipid NBs suspension was milk white. Under the light microscopy, lipid NBs and SonoVue MBs were small, spherical and distributed evenly (Figs. 1A and 1B). The mean size of lipid NBs was 565.6 ± 41.29 nm, and that of SonoVue MBs (SonoVue) was 2,429 ± 638.6 nm (Figs. 1C and 1D). The concentrations of lipid NBs and SonoVue MBs were (6.12 ± 0.62) × 109 bubbles/ml and (1.78 ± 0.22) × 108 bubbles/ml, respectively.

Figure 1 Characterization of lipid NBs and SonoVue MBs.

(A and B) Optical images of lipid NBs and SonoVue MBs in the microscope; (C and D) The size distribution of lipid NBs and SonoVue MBs by dynamic light scattering measurement.

Ex vivo HIFU synergistic effect assessment in excised bovine liver

After HIFU exposure, the ultrasound image of the targeted area appeared hyperecho (Fig. 2). Under the same HIFU parameters, the quantitative echo intensity of B-mode ultrasound image in lipid NBs groups was obviously higher than that in PBS groups (∗P < 0.05), but the quantitative echo intensity could not be demonstrated a difference between lipid NBs groups and SonoVue MBs groups (∗∗P > 0.05) (Fig. 4A). Meanwhile, the echo intensity of targeted area on B-mode ultrasound image after HIFU exposure was increased with the increasing of acoustic power. After HIFU exposure, the volume of necrotic tissues was also measured to compare the difference under diverse HIFU parameters (Figs. 3 and 4B). The coagulative necrosis volume between lipid NBs groups and SonoVue MBs groups could not be demonstrated a difference (∗∗P > 0.05), but the coagulative volume in lipid NBs groups was significantly larger than that in PBS groups (∗P < 0.05). These findings were consistent with the echo intensity results, indicating that lipid NBs could enhance the therapeutic efficiency of HIFU ablation comparable to that of SonoVue MBs.

Figure 2 Ultrasonoscopy of the targeted area in excised bovine liver before and after HIFU ablation.

(A1, B1, C1, D1, E1, F1, G1, H1, I1) Ultrasonoscopy of the targeted area in excised bovine liver before HIFU ablation. (A2, B2, C2, D2, E2, F2, G2, H2, I2) Ultrasonoscopy of the targeted area in the excised bovine liver after HIFU ablation with 200 µl PBS, SonoVue MBs and lipid NBs with concentration of 1.0 × 105 bubbles/ml at different acoustic power (120 W for 5 s; 150 W for 5 s and 180 W for 5 s). After HIFU ablation, the echo intensity of the targeted area (green mark) was significantly enhanced.

Figure 3 Photography of the targeted area in excised bovine liver after HIFU ablation.

After HIFU ablation, the targeted area showed grey white in color. The boundary against the surrounding tissue was sharp and clear.

Figure 4 Quantitative analysis of echo intensity and coagulative necrosis volume of the targeted area after HIFU ablation.

(A) Quantitative analysis of echo intensity and (B) quantitative analysis of coagulative necrosis volume of the targeted area in the excised bovine liver after HIFU ablation. ∗P < 0.05. ∗∗P > 0.05.

In vivo HIFU synergistic effect assessment in rabbit breast VX2 tumors

In this part, we further introduced lipid NBs and SonoVue MBs into rabbit breast VX2 tumor models to evaluate the synergistic effect of lipid NBs for HIFU thermal ablation compared with SonoVue MBs. When introducing lipid NBs and SonoVue MBs, the acoustic signal intensity of the targeted area showed obvious enhancement after HIFU exposure (Figs. 5A–5C). The quantitative echo intensity of B-mode ultrasound image showed that the lipid NBs did not show higher synergy compared to that of SonoVue MBs (∗∗P > 0.05), and dramatically higher than that of PBS (∗P < 0.05) (Fig. 7A). After HIFU exposure, the necrotic tissues were different with the ambient tissues on macroscopic inspection, showing that there was a well-defined boundary between them (Fig. 5D–5F). The necrotic tissues volume in in vivo tumor models, similarly, revealed that lipid NBs was not superior to that of SonoVue MBs (∗∗P > 0.05), but significantly larger than that of PBS (∗P < 0.05) (Fig. 7B). After HE staining, there was a sharp demarcation between ablated and non-ablated region in three groups (Figs. 6A–6C). In the all necrotic regions, the cells were seen with lysed cell membranes and nuclear fragmentation. In some necrotic regions of PBS group, some cells islets without HIFU ablation remained in the targeted area and arranged in nests (Fig. 6A green arrow). Visualizing by TEM, the structures of cells were ambiguous, and most of cell membranes and nuclear membranes were completely undefined in the three groups (Figs. 6D–6F). In the immunohistochemistry examination, the positive index (PI) of PCNA showed no difference between the lipid NBs and SonoVue MBs groups (∗∗P > 0.05), but the PI of PCNA in lipid NBs group was significantly lower than that in PBS group (∗P < 0.05) (Figs. 6G–6I and 7C).

Figure 5 Ultrasonoscopy and photography of the targeted area in the rabbit breast tumor after HIFU ablation.

Ultrasonoscopy showed echoes of the targeted area in rabbit breast tumor before (A1, B1, C1) and after (A2, B2, C2) HIFU ablation. (A) HIFU + PBS; (B) HIFU+ SonoVue MBs; (C) HIFU + lipid NBs. The green mark shows the range of tumor. (D, E, F) Photography of the targeted area in rabbit breast tumor after HIFU ablation combined with PBS, SonoVue MBs and lipid NBs, respectively. The necrotic tissue showed gray (white arrow) and non-ablated tissue appeared darkled (black arrow).

Figure 6 Pathological examination of the targeted area in rabbit breast tumor after HIFU ablation.

(A, B, C) HE staining of the targeted area after HIFU ablation (200 × magnification). A sharp demarcation was showed between ablated (white arrow) and non-ablated (black arrow) region. The green arrow showed the residual tumor cells in the targeted area in PBS group. (D, E, F) TEM photos of the targeted tissue after HIFU ablation. Cell membranes (black arrow) and nucear membranes (white arrow) were interrupted and indefinable. (G, H, I) Expression of PCNA in tumor tissue after HIFU ablation (200 × magnification). The brown (yellow arrow) indicated PCNA-positive cells, the blue indicated PCNA-negative cells (green arrow). (A, D, G) HIFU + PBS; (B, E, H) HIFU + SonoVue MBs; (C, F, I) HIFU + NBs.

Figure 7 Quantitative analysis of echo intensity, coagulative necrosis volume and PCNA positive index in tumor tissue after HIFU ablation.

(A) Quantitative analysis of echo intensity, (B) Quantitative analysis of coagulative necrosis volume and (C) PCNA positive index in tumor tissue after HIFU ablation in HIFU + PBS, HIFU + SonoVue MBs and HIFU + lipid NBs groups. ∗P < 0.05; ∗∗P > 0.05.

Discussion

The aim of this paper was to evaluate the synergistic effect of lipid NBs versus SonoVue MBs for HIFU thermal ablation in excised bovine liver and in vivo breast tumor models of rabbits. By carefully comparing the echo intensity of B-mode ultrasound images, the coagulative necrosis volume, and the pathological change after HIFU ablation in the presence of lipid NBs and SonoVue MBs both in ex vivo and in vivo experiments, these results showed that lipid NBs had the same effect as SonoVue MBs for synergistic HIFU thermal ablation. This further indicates that lipid NBs could potentially be used as an enhancer for synergistic HIFU thermal ablation of tumors.

HIFU focuses high-ultrasound-wave energy on the targeted region to produce a tremendous acoustic pressure, resulting in tissue necrosis due to thermal effect and cavitation effect (Farny, Glynn & Roy, 2010). However, bubbles in the targeted area could enhance the therapeutic efficiency of HIFU by accelerating ultrasound-mediated heating and lowering cavitation threshold (Tung et al., 2006; Kaneko et al., 2005). Bubbles in the high ultrasound pressure will experience nonlinear oscillation known as inertial cavitation, and radiate out of higher frequencies ultrasound with massive energy, which are more readily absorbed by tissues and availably converted into heat to damage the tumor cells (Umemura, Kawabata & Sasaki, 2005; Holt & Roy, 2001). Therefore, bubbles are an important enhancer for HIFU thermal ablation of tumors. Sokka, King & Hynynen (2003) had ever studied endogenous bubbles directly from the tissue to enhance the ultrasound absorption and ultimately create larger lesions in vivo, but this required very high acoustic power and the number and activity of the resultant bubbles was highly variable due to the heterogeneity of tissues. Therefore, exogenous bubbles are potentially an ideal alternative to lower the cavitation threshold, enhance the cavitation activity, and improve the therapeutic efficiency of HIFU thermal ablation.

MBs are well known to be an important enhancer for HIFU thermal ablation by ultrasound-mediated tissue heating and cavitation effect. However, the clinical translation of MBs as an ablation enhancer in HIFU treating tumors is basically limited by its disadvantages. Firstly, MBs are too large to extravasate out of tumor vascular space. Secondly, MBs have a very short blood circulation time. Additionally, excess MBs easily shift the heating position from the targeted area and cause unwanted heating and irreversible thermal damage to healthy tissue (Moyer et al., 2015). With the rapid development of nanotechnology, NBs with various shells present a promising application in disease diagnosis and treatment due to its good imaging ability, long lifetime in blood circulation and strong infiltration out of the endothelial gap of tumors (Yin et al., 2012; Fan et al., 2013). In this study, we compared the synergistic effect of lipid NBs and SonoVue MBs for HIFU thermal ablation of tumor. Between lipid NBs groups and SonoVue MBs groups, the echo intensity within region of interest on B-mode ultrasound images was significantly larger than that of corresponding PBS groups. However, there was not a significant difference between lipid NBs and SonoVue MBs groups both ex vivo and in vivo experiments. In order to further investigate the therapeutic effect of bubbles for enhancing HIFU thermal ablation, we evaluated the coagulative necrosis volume and pathological change after HIFU thermal ablation. These results showed that lipid NBs had the same synergistic effect as SonoVue MBs during HIFU thermal ablation process. Therefore, lipid NBs could potentially become an ideal enhancer of HIFU ablation of tumors.

After HIFU exposure, the appearance of hyperecho of B-mode image at HIFU focus was commonly used to estimate coagulative necrosis of tissue and the volume of necrosis during US-guided high intensity focused ultrasound treatment procedure. However, the mechanism of appearance of hyperecho was currently unclear. Rabkin, Zderic & Vaezy (2005) believe that the onset of cavitation had a strong correlation with the appearance of hyperecho at HIFU focus. Their passive cavitation detection results showed that inertial cavitation occurred prior to the appearance of a hyperechoic region on B-mode ultrasound image. Coussios et al. (2007) believe that the appearance of hyperechoic regions on B-mode ultrasound image constituted neither a necessary nor a sufficient condition for inertial activity to occur during HIFU exposure, but boiling cavities played a significant role in monitoring HIFU therapy as they were readily visible on B-mode ultrasound image. In our paper, the echo intensity on B-mode ultrasound image between lipid NBs and SonoVue MBs groups was significantly higher than that of corresponding PBS groups, but it could not demonstrate a difference between lipid NBs and SonoVue MBs groups. We speculated that the hyperecho region was correlated with the production of a mass of bubbles induced by cavitation activity and boiling. Simultaneously, the infusion of lipid NBs or SonVue MBs provided extra bubbles at HIFU focus, which contributed to the production of cavitation bubbles or boiling bubbles. Because of complex and unpredictable of the behavior of acoustic cavitation, the mechanism of between hyperecho on B-mode ultrasound image and acoustic cavitation or boiling need be further studied.

When exposed at high ultrasound pressure, gas-filled bubbles exhibit different destruction mechanisms to biological tissues. Apart from the thermal effect and mechanical action, shock waves, high fluid velocities and free radicals from cavitation also play an important role in lesion formation. When remarkable cavitation is induced in situ, the generated bubbles potentially act as ultrasound scatters and increase ultrasound power deposition in targeted area. Recently, many means were investigated to enhance local heating and cavitation activity during HIFU ablation. The varying components of bubbles’ shell membrane and different types of substances inside bubbles have potentially exhibited different efficiencies in tissue heating and cavitation activity. Zhang et al. (2012) confirmed lipid-shelled MBs had a greater efficiency than polymer-shelled MBs in heating and cavitation during focused exposures. Compared to the hard-shelled polymer MBs, the soft-shelled lipid MBs could easily lead to higher harmonics that are more readily absorbed and converted to heating deposition in the targeted area by nonlinear oscillations. However, the small NBs have higher resonant frequency than MBs in the same acoustic field. Whether the small NBs could induce larger lesions than MBs needs further studies using different shelled bubbles and different size of bubbles at high ultrasound pressure. Zhou et al. (2015) used uSPIO/PLGA nanoparticles as contrast agents for the enhancement of the effects of HIFU ablation on liver tissue. The uSPIO nanoparticles in the shell of the microspheres could boost the acoustic impedance to generate stronger ultrasound scattering and improve heating deposition in the targeted area. Compared to uSPIO nanoparticles, gas-filled bubbles could exhibit stronger acoustic impedance difference between bubbles and surrounding biological tissues, which more easily lead to heating deposition in the targeted area. The shelled components of NBs used in this work were similar with that of SonoVue MBs, and the gas core was different. Thus, more studies will be needed to elucidate the influence of components of bubbles in high pressure ultrasound field and promote the clinical translation.

In addition to enlarging the tissue lesions, bubbles could reduce the occurrence of residual tumor during HIFU thermal ablation. Pathological inspections in this paper revealed that there were no non-ablated cells within the targeted area in both lipid NBs group and SonoVue MBs group, but one observation worth noting is that there were still some non-ablated cells within some targeted area in the PBS group (Fig. 6A green arrow). These findings have been reported in some previous literatures (Orsi et al., 2015; Boutier et al., 2011). This study also showed the synergistic therapy of bubbles for HIFU thermal ablation can effectively reduce the residual tumor in the targeted area.

The thermal effect and cavitation activity play an important direct role in killing tumor cells during HIFU ablation. Recently, however some studies showed that HIFU also has a potential to induce the whole body antitumor immune response for effective tumor therapy (Unga & Hashida, 2014). HIFU destruction of tumors may lead to immunity forming in the body by infiltration of immune cells into the tumor and exposure of antigen. Some scholars demonstrated that the formation of cell debris generated by cavitation activity and mechanical effect of HIFU was more beneficial to activation of the whole body antitumor immune response, rather than coagulation necrosis come from thermal effect of HIFU (Hu et al., 2007; Wu et al., 2004). In our immunohistochemical examination, the positive index of PCNA was obviously reduced, especially in both lipid NBs group and SonoVue MBs group. It well reflected that HIFU exposure contributed to suppress the proliferation of tumor cells, particularly in the presence of bubbles. In addition, lysed cell membranes and nuclear fragmentation contributed to activate the immune response. But we did not try to detect and analyze the antitumor immune response in the present study. Therefore, the exact mechanism of antitumor immune response induced by bubbles in the acoustic field needs further investigation.

Several limitations of our study should be addressed. First, the biocompatibility, biodistribution and biosecurity of lipid NBs in vivo are not shown in our study. Second, we did not explore the temperature change caused by the bubbles in this study. Third, we did not directly investigate the enhanced permeability and retention (EPR) effect of NBs in the tumor. Finally, we just employed single “ablated-dot” mode in HIFU thermal ablation of rabbit breast VX2 tumors. The effect of the whole tumor ablation by HIFU combined with lipid NBs should be further addressed.

Conclusions

In this study, we introduced lipid NBs and SonoVue MBs into the targeted area of HIFU thermal ablation and explored the synergistic effect of lipid NBs and SonoVue MBs for HIFU thermal ablation in ex vivo bovine liver and in vivo breast tumor models of rabbits. By analysis of the echo intensity change of B-mode image after HIFU thermal ablation, the volume of necrotic tissues, macroscopic and microscopic examinations of necrotic tissue and immunohistochemical examination of PCNA of tumor cells, these results showed that lipid NBs had the same effect as SonoVue MBs for synergistic HIFU thermal ablation. All in all, our study suggested that lipid NBs had the same effect as SonoVue MBs for synergistic HIFU thermal ablation with similar shell materials. In conclusion, lipid NBs are not only a good contrast agent for ultrasound molecular imaging and a fine vector for drug delivery and gene transfection, but also an potentially enhancer for HIFU thermal ablation of tumors.

Supplemental Information

Data S1 NBs and MBs

Click here for additional data file.

Data S2 Excised Bovine liver

Click here for additional data file.

Data S3 Rabbit breast tumor

Click here for additional data file.

We would like to thank Chongyan Li for providing the VX2 tumor tissue, and Dan Zhou and Fenfen Xu for performing excised bovine liver experiment.

Additional Information and Declarations

Competing Interests

Author Contributions

Animal Ethics

Data Availability

The authors declare there are no completing interests.

Yuanzhi Yao conceived and designed the experiments, performed the experiments, analyzed the data, contributed reagents/materials/analysis tools, wrote the paper, prepared figures and/or tables, reviewed drafts of the paper.

Ke Yang conceived and designed the experiments, performed the experiments, contributed reagents/materials/analysis tools, reviewed drafts of the paper.

Yang Cao analyzed the data, contributed reagents/materials/analysis tools, reviewed drafts of the paper.

Xuan Zhou, Jinshun Xu, jianxin Liu and Qi Wang performed the experiments, contributed reagents/materials/analysis tools.

Zhigang Wang conceived and designed the experiments, contributed reagents/materials/analysis tools, reviewed drafts of the paper.

Dong Wang conceived and designed the experiments, analyzed the data, contributed reagents/materials/analysis tools, reviewed drafts of the paper.

The following information was supplied relating to ethical approvals (i.e., approving body and any reference numbers):

The Animal Ethics Committee of Chongqing Medical University (SYXK (Chongqing) 2012-0001).

The following information was supplied regarding data availability:

Raw data can be found in the Supplemental Information.

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
