# Peer review of "Comparison of the synergistic effect of lipid nanobubbles and SonoVue microbubbles for high intensity focused ultrasound thermal ablation of tumors"

_PeerJ, doi:10.7717/peerj.1716_

## Round 0.1 · original submission · Major Revisions

The paper describes an interesting experiment comparing NanoBubbles and MicroBubbles for their effectiveness in HIFU ultrasound therapeutic purposes. The reviewers provided fair and rigorous feedback that should provide ample material to substantially improve the manuscript. In addition I'd like to mention the unclear PBS abbreviation in the abstract. Furthermore, the NB contrast reference is to "Wand et al. 2012b" which is not in the reference list. Presumably a typo.

Reviewer 1 ·

Basic reporting

The article, in its present form, is in need of extensive editing and proof-reading for English spelling and grammar. Although, for the most part, it is relatively clear what the author was trying to convey, I think the written language should be improved before this paper is accepted.
Figures that show ultrasound images (Fig.5 and Fig.2) would be more informative if the targeted region was cropped out and enlarged/zoomed in. Otherwise, it is very hard to appreciate the differences in hyperechoic changes between different power settings. Also, it would be great to remove the green mark that shows the target, if possible, because it obscures the changes in the image.

Experimental design

The authors should comment on how the three different exposure conditions were selected. Why were the output powers set at 120, 150 and 180 W? Is it known what the peak focal pressures are at these outputs? And why the exposure durations chosen at 5 s?
In the in vivo experiments, was the presence of NBs in the tumor visualized by ultrasound imaging? How was the NBs dose determined? And why was the HIFU exposure applied 15 s after the administration of NBs? My understanding was that the very idea of using NBs for tumor imaging/therapy is that they get accumulated in the tumor through EPR effect. Is the 15s time gap sufficient to allow accumulation? Please comment.

Validity of the findings

The conclusion "Thus, NBs will become a promising enhancer for synergistic
HIFU ablation." is an overstatement. The authors have shown in these studies that the NBs enhance the HIFU heating effect as well as the MBs in a superficial target. However, the issue with using NBs (as well as MBs) in clinical HIFU ablations with systemic administration is that enhanced heating, as well as enhanced mechanical effects are induced not only at the focus, but prefocally as well, in all the intervening tissues containing bubbles. Cavitation threshold is dropped in the intervening tissues by just as much as it is dropped at the target. In fact, Tung et al. (Tung YS, Liu HL, Wu CC, Ju KC, Chen WS, Lin WL. Contrast-agent-enhanced ultrasound thermal ablation. Ultrasound Med Biol. 2006 Jul;32(7):1103-10.) have directly demonstrated that, in the presence of ultrasound contrast agents, the lesion position shifts, and cavitation occurs well outside of the targeted region. The authors should at least mention these implications in the Discussion section.

Reviewer 2 ·

Basic reporting

Language: The English used in the manuscript need to be improved. My suggestion would be 1- to make shorter sentences and 2- to ask an English native speaker to help reviewing the English.
Abstract: headings are missing
Line 98-99 DppC and DPPE are not defined
Line 226-230 and line250-256 : sentences need to be improved, the meaning is not clear.
Technical language:
In the title and across the manuscript, the expression ‘‘ultrasound ablation’’ is used. Different types of ultrasound ablation exist, like histotripsy, boiling histotripsy, thermal. To my understanding, the authors are using ultrasound thermal ablation. Authors should make it clear by adding the word thermal to ablation.
HIFU exposure parameters as power had been used in the article. Which kind of power, electrical or acoustic? It should be clarified from the first time it appears in the manuscript.
Line 127: Ex vivo would be more appropriated than in vitro, same comment where in vitro is appearing in the manuscript for bovine liver experiments.
Title and in the article: description and meaning of the synergistic HIFU thermal ablation need to be explain in the introduction.
Word ‘gray-scale’ is used everywhere in the article. What is it? It is not the correct word to use. Could it be B-mode image? Ultrasound pulse inversion image? MRI images? Please change the word to gray scale of the … MRI image or B-mode image or …. to make it clear.

Literature:
The introduction is not covering the background of the study and need to be rewrite and improved:
Line 45: FDA approval please provide a reference,
Line 53-54 HIFU limitation and treatment duration. Missing references: about multi-element transducer and electrical steering, about bubble enhanced heating without the need of injection of cavitation nuclei in the bloodstream, and boiling histotripsy. drug delivery.
What would be advantage of using smaller acoustic cavitation nuclei in HIFU ablation treatment (thermal or not)? This is missing in the introduction.
Anti tumour effect only appears in the discussion. State of art of this finding should appear in the introduction.
What is PCNA technique; please provide a reference and a full description of the methods in the materials and methods section
A similar study from Di Zhou et al. in RSC advances 2015 show similar results with different particle. This study should be cited and used if possible for comparison.
The authors cited studies about nanodroplets used to enhanced the HIFU ablation. How nanodroplets differs from your NBs? That should be explained in the introduction.
The state of art of non-thermal effect of HIFU and bubble is missing: hystotripsy, acoustic cavitation,…


Figures:
Figure2 and 5: more description of the image is required. What is visible on the image? What is the green region? And the scale?
Figure 4: what is ‘mean change of the gray scale’? Does it have any units? Please provide the description of the calculation in the legend or better in the materials and methods? And again what is the gray scale??

Experimental design

Nanobubbles preparation:
Nbs life duration in a storage vial and/or in the bloodstream had not been described. Biocompatibility of NBs is also not in the article. Please provide the results if it is available.
Please make clear for each experiments the NBs and MBs concentration and volume that was used.

Animal preparation:
Reference for animal model tumour VX2

Method missing about the sacrifice procedure of animals.
Ultrasound equipment:
Description of the ultrasound setup is not complete. Maybe a dedicated section in the materials and methods could help to address the lake of information. What the pressure at the focus with the different acoustic powers used in the study? Is the ablation made in continuous mode or pulsed mode?
Is there an ultrasound imaging probe (a figure of the experimental setup could help to understand), and what are probe parameters (central frequency, number of elements, aperture)?
How ultrasound images data had been processed? Please add a description. Is the mean change of the gray-scale of the image is taken in a region of interest? on the last frame?, then each pixels summed and divided by the number of pixel? How is defined the region of interest? How many pixels ? Spatial resolution of the image?

Ex vivo bovine liver experiments:
How many samples had been tested? Please make it clear in the text or by creating a table.
Is any measurement or simulation of the temperature had been performed at the focus?
In vivo experiments:
Please clarify the concentration of NBs and MBs for this part.
Why waiting 3 days before taking of the tumours?
Histology examinations:
It is not clear if the same sample had been used for macrospic examination, then for histology, then for TEM, then for immunohistochemical examination or if one sample was used for one method.
Please add a description of PCNA procedure.

Validity of the findings

The use of ‘ultrasound ablation’ and missing parameters about the HIFU exposure make ambiguous the title and the conclusion. To my understanding but I may be wrong, the HIFU treatment was in continuous mode. In their experiments, authors showed that no significant differences could be observed in term of lesion, PCNA, or ultrasound images by changing the size of the bubbles. Results could be very different in pulsed mode and are not investigated in this study. Consequently the conclusion and the title should be change according to the parameters of the study.
In the abstract, the conclusion was ‘’the bubble size was not critical factor for synergistic HIFU ablation’’.
Authors should be careful with such statement. Results do not fully support that conclusion. Firstly, chemical composition of MBs and NBs is different: a different gas is used and also different components for the shell. Consequently, biological and chemical impact of MBs and NBs could be different. Authors should discuss it in the discussion part. Conclusion sentence in the abstract should be rephrased.
Line 227-230: Please clarify sentences and rephrase.
Line 246-247 : ‘’At a lower acoustic pressure, the inertial cavitation thresholds of bubbles decrease with increasing the frequency and pressure of the ultrasound wave.’’ That statement is wrong. The sentence need to be rephrased or removed.
Line 243-253. Please clarify sentences and rephrase.
Di Zhou et al. had published study in 2015 in RSC advances 2015. Their experimental protocol and equipment seems to me very similar, as well as their method to quantify changes in the gray-scale of the ultrasound images. Authors should compare their results with already published studies. Di Zhou et al. is just one example.
A difference in echogenicity on ultrasound images is used as a monitor of the effect. My question is why?
Authors wrote in the discussion part: ‘’Based on previous literature reported (Chen, Li & Wan, 2006), the change of echo, appearing hyperecho after HIFU ablation, was due to the generation of bubbles in the targeted area in the process of HIFU ablation, believing that the larger bubbles were newly generated from the coalescence of collapse of preceding bubbles at high ultrasound pressure of HIFU. ‘’ The article written by Chen, Li & Wan, 2006 showed the inception of cavitation bubble clouds induced by HIFU in water by using a high speed camera. No ultrasound images were performed. Consequently this study is not supporting authors statement. Please rephrase according to the literature. Since no temperature measurement had been described, authors should also consider boiling as a potential candidate for echogenicity in the ultrasound images. In some studies, it had been reported that shell composition also played a role in the echogenicity that could be observed in ultrasound images. That is missing from the discussion.

---

## Round 0.2 · Minor Revisions

Please properly address the valid comments of reviewer 2. I totally agree that English spelling is still a problem in the manuscript. There are simple spelling mistakes that any word processor can find and correct for you. Please have an English speaker read and correct the manuscript.

A few of my own observations:
* What is Gray Val 1.0 Software, what do you measure? I do not understand the purpose of the sentence: "Meanwhile, the echo intensity change was increased with the increasing of acoustic power." in the results. If intensity can vary between measurements how do you ensure the differences measured are not due to these variations?
* Poor English throughout the document. For example: "was no significant difference" should be "was not significantly different". Again find somebody able to correct these basic English mistakes.
* Stating something is similar (**p>0.05) is a statistical error. Rephrase to "could not demonstrate a difference". Because, maybe if your dataset was larger you could demonstrate significance.

Reviewer 1 ·

Basic reporting

No comments. My previous comments have been addressed and I have no further questions/comments.

Experimental design

No comments. My previous comments have been addressed and I have no further questions/comments.

Validity of the findings

No comments. My previous comments have been addressed and I have no further questions/comments.

Additional comments

No comments. My previous comments have been addressed and I have no further questions/comments.

Reviewer 2 ·

Basic reporting

Language: There is still a lots of mistakes. However, from my point of view it was understandable.
Line 144 and 146 there are two almost identical sentences. Please correct it.

Technical language:
Vocabulary changes were made from ablation to thermal ablation in the title and in the manuscript. However, some of them were forgotten. Please correct it.

Literature:

HIFU limitation and treatment duration.
Some references have been added by authors but with very poor description. Please improve it.
To my understanding, NBs or MBs are used in the article to enhance heating at the focus. It is not the only mechanism that is happening in presence of bubbles at the focus. Very short HIFU pulse with high to moderate amplitude can lead to tissue ablation in absence of any temperature rise. Is immune response is also stimulated in non thermal cases? Which kind of biological effects were observed? Is there any similarity with biological effect observed in your study?

HIFU ablation/treatment of large volume of tissue is a limitation due the small size of the focus compared to the size of treated area. The size of focus is limited by the geometry of the transducer when it is a single element. Methods based on multiple elements transducer (fast electronic steering, focus with a ring shape,… ) demonstrated interesting results to overcome this limitation and is not presented in the introduction.

Experimental design

Ultrasound equipment:
Figure 1, added to describe the experimental setup, is useless. Please improve it by adding setup parameters (focal length, frequencies,…) that have been added in the material and methods

In vivo experiments:
In ex vivo tissues, thermal lesions are visible just after the exposure. In vivo authors justify a waiting time of three days because lesions appeared to be more visible according to preliminary experiments. What are biological mechanisms responsible for that? Have you checked after one month? Two months? Please also add it to the manuscript.

Validity of the findings

Authors described in their answer that the hyperecho observed on B-mode image is due to ‘a mass of bubbles’.
Please added it to the manuscript.
In the manuscript authors did not clearly explain the origin of this bubbles. However, only one bubble mechanism was described in the manuscript: acoustic cavitation. In absence of any temperature measurement, or simulations and with such intensity at the focus, boiling may probably be involved in the process. Literature reported that when boiling occurred a hyperecho appeared on b-mode images.
Please make it clear in the manuscript.

---

## Round 0.3 · accepted · Accept

Comments have been addressed appropriately. This paper adds to the knowledge base that nano-bubbles can act as enhancer for HIFU therapy. I am confident this paper will contribute to the understanding of this interesting research field.